# Crypto-coding technique based on polar code and secret key generated from wireless channel characteristics for wireless communication systems

**Dinh Van Linh[1,2], Vu Van Yem[1], Hoang Thi Phuong Thao**📙[3]*

**1** Hanoi University of Science and Technology, Hanoi, Vietnam, **2** Academy of Cryptography Techniques, Hanoi, Vietnam, **3** Electric Power University, Hanoi, Vietnam

* thaohp@epu.edu.vn

**Data availability statement:** All relevant data are within the manuscript and its Supporting Information files.

## Abstract

The crypto-coding technique is essential for modern digital wireless communications systems, allowing data encryption and channel coding to be performed in only one step without requiring additional hardware. This work proposes a crypto-coding technique combining polar codes with a secret key, which is derived from wireless channel characteristics, to boost the security and reliability characteristics of the systems. The secret key is divided into two parts, one is assigned to the frozen bits, and the other is XORed with the information bits. A simulation with different key lengths and code lengths is carried out in Additive white Gaussian noise (AWGN) that helps us to evaluate this technique by the error correction and security performance, computational complexity. The key benefit is that it achieves the same bit error rate (BER) performance and computational complexity as traditional polar codes and existing methods despite only taking one step. Meanwhile, it ensures completely degraded decoding effectiveness at eavesdroppers, thus it is effective against passive eavesdropping attacks. Furthermore, this method does not require additional hardware overhead for key management and distribution.

## I. Introduction

Security and reliability are crucial factors for modern wireless communication systems [1]. They can be implemented at the physical and the upper layers which complement each other to ensure secure and reliable communication in systems. In terms of reliability, the upper layers use error correction and retransmission techniques, whereas the physical layer provides an important contribution to ensure accurate data delivery, which can employed by channel coding [2]. For security, they offer robust encryption mechanisms, but at a higher resource cost. In contrast, the physical layer security protects data with a lower resource consumption by relying on the physical characteristics of the communication channel [3]. Therefore, physical layer security has been receiving more attention from researchers [4]. Traditionally, data encryption and channel coding at the physical layer are two completely independent blocks

**Funding:** Dinh Van Linh was funded by the Master, PhD Scholarship Programme of Vingroup Innovation Foundation (VINIF), grant no. VINIF.2023.TS.057.

**Competing interests:** No authors have competing interests.

in most secure wireless communication systems, which increases the computational complexity and latency [5]. To overcome these disadvantages, the crypto-coding techniques that performs encryption and error correction in a single step has attracted significant attention in the last decade. Previous studies proposed crypto-coding technique based on various channel coding such as low-density parity-check (LDPC) code for mobile 5G [6], Turbo code for software-defined radio systems, satellite and mobile (3G, 4G) [7–17], and polar code for 5G [18–30]. Polar codes, found by Arikan in 2009, are primarily utilized as channel code in 5G thanks to their low encoding and decoding complexity, high error correction performance, and flexible code rate [31]. The crypto-coding technique is performed by using secret keys, which can be generated from classical methods [18–25] or wireless channel characteristics [26–30], to control either the frozen bits or the information bits of polar code. However, the techniques using the secret keys generated from classical methods lead to an increase in computational complexity and require a complicated key management infrastructure. On the other hand, key generation from wireless channel characteristics is a potential alternative because of its lightweight and requirement of simpler hardware [32]. Therefore, the combination of polar codes and keys generated from wireless channel characteristics is a promising technology in the 5G system. In [26], the authors use a secret key extracted from channel state information (CSI) to assign frozen bits of polar code. The proposed technique provides the same bit error rate (BER) as the traditional method and a large key space. The work in [27] proposes a technique to improve physical layer security by using random channel gain of legal transmission to find out the frozen bit patterns. The investigation reported in [28], a frozen-bit pattern algorithm based on a channel quality indicator is developed to encrypt the transmitted message over the wireless channel, leading to scenarios a low block error rate (BLER). The authors in [29] measure channel entropy in real scenario to extract the secret key. Then, the secret key is assigned to the frozen bits of the polar code. In [30], the authors use the random phases extracted from wireless channels as an input chaotic sequence generator. To improve the security level, the authors add the Advanced Encryption Standard (AES) block to encrypt the generated chaotic sequence. The investigations in [26–30] have evaluated the reliability and security level by analyzing BER/BLER, or signal-to-noise ration (SNR). However, these studies are limited in evaluating the error correction efficiency, computational complexity, and security level. Especially, they lack security evaluation according to the test of National Institute of Standards and Technology (NIST) 800-22REV1A. Furthermore, it is worth noting that only a few studies on polar-based crypto-coding techniques using secret keys generated from wireless channel characteristics have been published in the open literature. Therefore, we believe this is an interesting topic to explore and a promising candidate to significantly enhance the reliability and security of 5G systems.

In this paper, we propose a crypto-coding technique based on polar code and secret key. The secret key is extracted from the channel impulse response (CIR) of the transmitter (Alice) – legal receiver (Bob)'s link in time division duplexing (TDD) mode. The secret key, which has the same length as the polar code, is divided into two parts. The bits in the first part and the second part have the same indices as the bits in the frozen and information patterns, respectively. Then, the bits in the first part are assigned to the frozen bits, while the bits in the second part are XORed with information bits. In this way, eavesdroppers without the generated secret key cannot decode and decrypt the transmitted data. The proposed method is performed through the Additive White Gaussian Noise (AWGN) channel with various code lengths and code rates. To prove the effectiveness of error correction, the BER of the system using the proposed method will be compared to that using traditional polar codes. Simultaneously, security effectiveness is also evaluated by the BER, which is compared between Bob's and Eve's decoding processes based on the BER similarity percentages between Bob and Eve.

Eve's BER is completely different from Bob's BER, which proves that Eve is completely unable to accurately intercept and receive the information exchanged between Alice and Bob. As a result, the systems obtain perfect secrecy. In summary, the contributions of this research are described below:

- The secret key generated from CIR ensures randomness, according to National Institute of Standards and Technology (NIST) 800-22REV1A. This key generation method is lightweight and has low complexity, with no need for a third party to manage and distribute keys.
- The frozen bits and information bits of the proposed secure polar code are dynamically controlled by a single secret key. This reduces the complexity of encryption and decryption processes.

The key generation method used in this paper is presented in detail in our study [33]. The secret keys with lengths of 128 bits, 256 bits, 512 bits, 1024 bits, and 2048 bits are generated from the CIR of the systems. Next, the randomness of these secret keys is assessed by NIST 800-22REV1A before applying them to our proposed secure polar code.

The remaining sections of this paper are as follows. Section II shows materials and methods. Section III gives results and discussion. Finally, Section IV concludes this research.

## II. Materials and methods

### Secret key generation from wireless channel characteristics

We consider a system model with three nodes, where Alice and Bob are legal parties and Eve is an eavesdropper, as displayed in Fig 1. In our scenario, three nodes have the same configuration. $h_{AB}$ and $h_{BA}$ are CIRs of Alice-Bob's link and Bob-Alice's link, respectively. Meanwhile, Eve obtains Alice and Bob via the wire-tap channels $h_{EA}$ and $h_{EB}$.

Fig 2 displays the flow chart of key generation for the systems using the proposed method in [33]. Due to reciprocal properties between legitimate nodes, Alice and Bob obtain the same CIR in TDD mode with coherence time ($h_{AB} = h_{BA}$). Eve stands at a distance greater than half a wavelength from the legitimate users, so there is independence between the main channel and the wire-tap channel [34]. As a result, Eve is unable to learn any CIRs about Alice and Bob. In our research, Alice and Bob select the same maximum peaks from the estimated CIRs after the channel probing step. They then apply the quantization function $f(\cdot)$ in [33] to extract their secret keys named $K_A$ for Alice and $K_B$ for Bob.

$$\begin{cases} K_A = f(h_{AB}) \\ K_B = f(h_{BA}) \end{cases} \tag{1}$$

Alice and Bob obtain the same secret keys ($K_A = K_B$). Using the same approach, Eve also employs this quantization function $f(\cdot)$ to estimate his secret keys from $h_{EA}$ and $h_{EB}$.

$$\begin{cases} K_{EA} = f(h_{EA}) \\ K_{EB} = f(h_{EB}) \end{cases} \tag{2}$$

The wire-tap channel has channel statistics different from the main channel, so Eve cannot extract the correct secret key. After each communication session, Alice and Bob continuously update the secret keys instead of relying on the previous keys. As a result, this key extraction method improves the secret key's security because it is difficult to predict the key immediately.

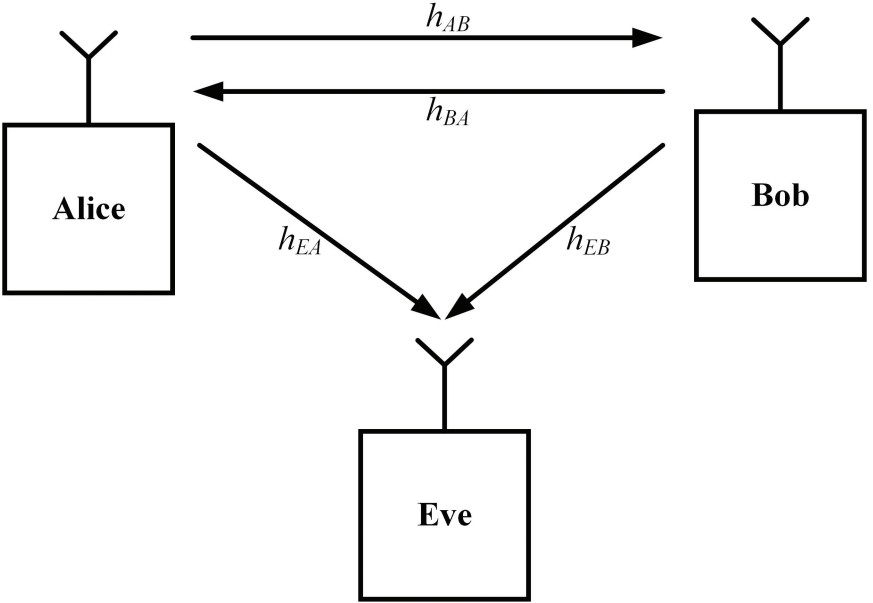

**Fig 1. The system model.**

## Polar code

Polar codes are the channel codes that utilize channel polarization to achieve the Shannon limit of a Binary-input Discrete Memoryless Channel (B-DMC). Channel polarization is obtained by repeatedly executing the polarization transformation. The encoding and decoding principles of polar code can be explained as the following.

Given the code length $N = 2^n$, $n = 1, 2, \ldots$, the binary source block $u_1^n = (u_1, u_2, \ldots, u_N)$ consists of $P$ information bits and $(N–P)$ frozen bits $u_{A^c}$.

The codeword $x_1^N$ with a code rate $R = P/N$ is acquired as follows:

$$x_1^N = u_1^N G_N \tag{3}$$

where the generator matrix $G_N$ can be expressed as:

$$G_N = B_N F^{\otimes log_2 N} \tag{4}$$

$B_N$ is the permutation matrix, $F = \begin{bmatrix} 1 & 0 \\ 1 & 1 \end{bmatrix}$, and $\otimes$ is the inner product [31].

$A$ is a $P$-element subset of $\{1, 2, \ldots, N\}$. $A$ is encoded to obtain $u_A G_N(A)$, while $u_{A^c} G_N(A^c)$ is achieved after encoding the frozen bits $u_{A^c}$. The codeword in Eq (3) can be rewritten as below:

$$x_1^N = u_A G_N(A) + u_{A^c} G_N(A^c) \tag{5}$$

The codeword $x_1^N$ is transmitted through the channel $W^N$ and then received $y_1^N$ at the channel output. Since the frozen bits of the decoder $\hat{u}_{A^c}$ are assigned the same as $u_{A^c}$, the decoder's function is to produce an estimate value $\hat{u}_1^N$ of $u_1^N$.

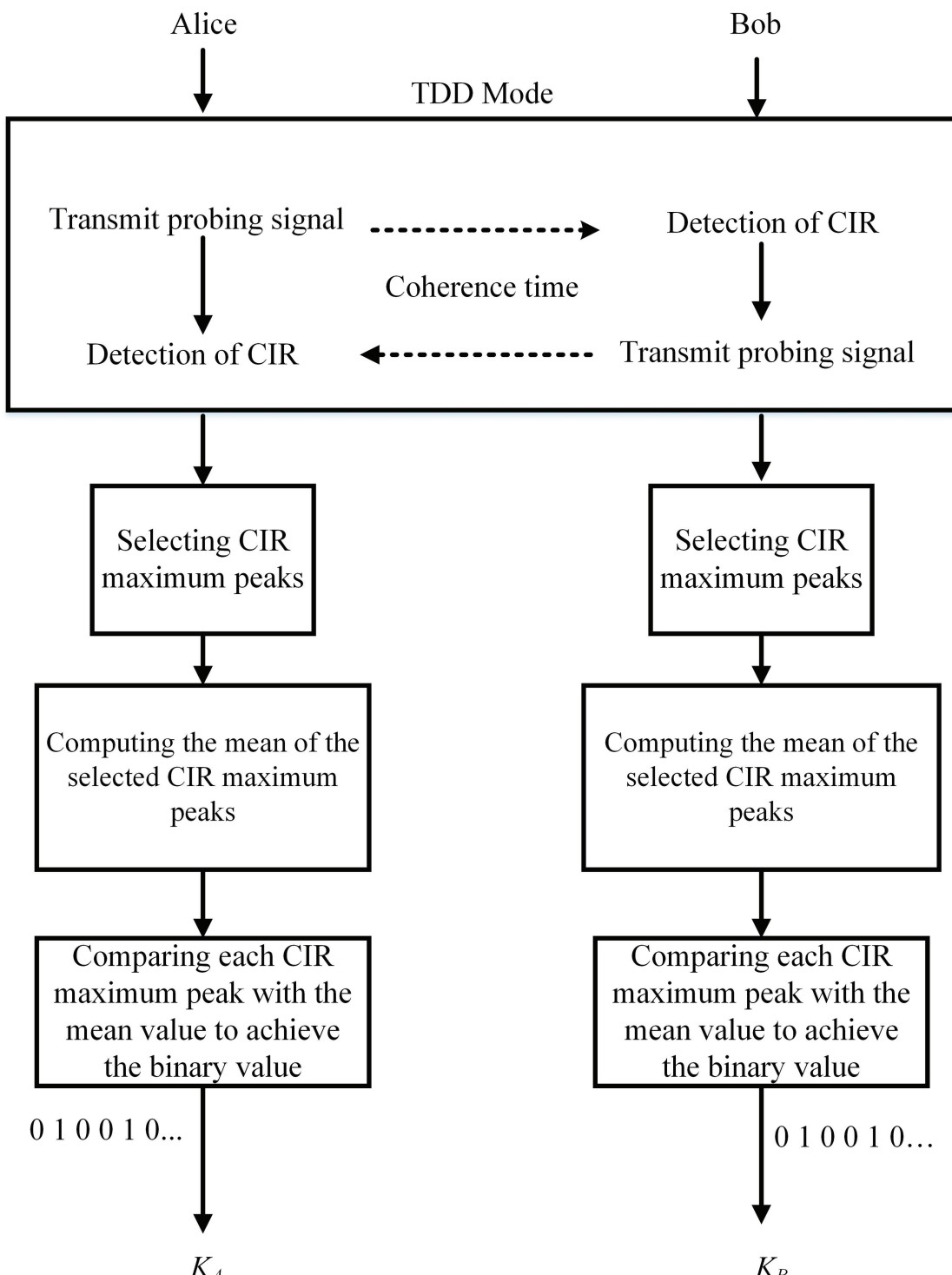

**Fig 2. Secret key generation steps for the wireless communication systems.**

A successive cancellation (SC) decoder applies a hard decision approach to the decoding principle by iteratively calculating the bit channel's likelihood ratio (LR) information.

SC decoder generates the estimate $\hat{u}_1^N$ by the decision rule as follows:

$$\hat{u}_i \triangleq \begin{cases} u_i, & \text{if } i \in A^c \\ h_i\left(y_1^N, \hat{u}_1^{i-1}\right), & \text{if } i \in A \end{cases} \tag{6}$$

where $i = 1, 2, ..., N$ and $\{u_i, i \in A^c\}$ is the frozen bit. The decision function is determined as:

$$h_i\left(y_1^N, \hat{u}_1^{i-1}\right) \triangleq \begin{cases} 0, & L_N^i\left(y_1^N, \hat{u}_1^{i-1}\right) \geq 1 \\ 1, & \text{otherwise} \end{cases} \tag{7}$$

The main task of the decoder is to compute the bit channel's LR and transition probability.

### Proposed secure polar code

The secure transmission model is displayed in Fig 3. Fig 3 shows that Alice transmits a secret message to Bob while Eve attempts to receive it. Alice and Bob distill the same secret key for controlling the proposed secure polar code.

The proposed method has a size of $N$ bits with $P$ information bits $u_A$ and $(N-P)$ frozen bits $u_{A^c}$ which creates a data source vector $u_1^N = (u_A, u_{A^c}) = \{u_1, u_2, ..., u_{N-1}, u_N\}$. The information and frozen bit positions $(A/A^c)$ are synchronized at both the transmitter and receiver. In data encryption, the length of the secret key must match the size of the plaintext block [35]. Therefore, we generate the secret key with a length of $N$ bits in our proposed method. The secret key will then be divided into two parts, including $K_1$ and $K_2$, which have the sizes of $P$

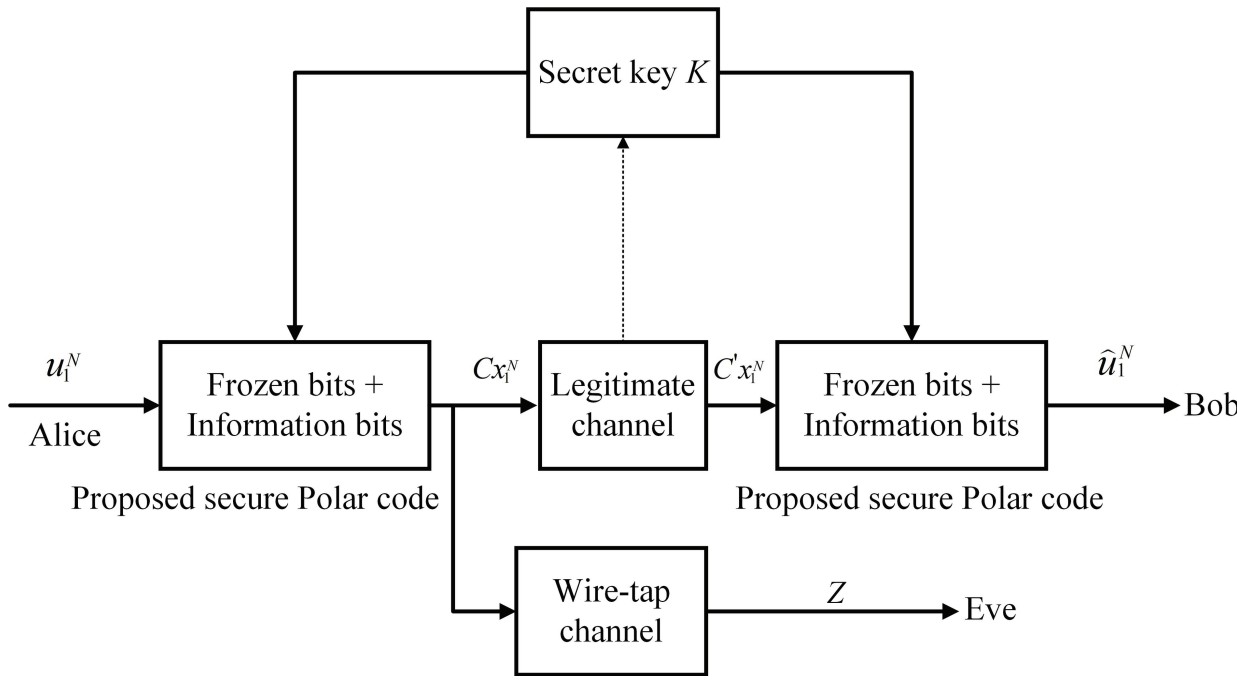

**Fig 3. The secure transmission model based on the polar code.**

and $(N-P)$ bits, respectively. The bit positions of $K_1$ and $K_2$ are the same as the bit positions of information and frozen parts, respectively.

Fig 4 displays the proposed secure polar code. The first secret key part $K_1$ is XORed with the information bits $u_A$ to obtain $u'_A$.

$$u'_A = u_A \oplus K_1 \tag{8}$$

For the conventional polar code, the frozen bits are wasted and set up by zero sequences. In this proposed method, the second secret key part $K_2$ is assigned in the frozen bits $u_{A^c}$. This makes it impossible for attackers to guess the frozen values. At this time, the frozen bits will be:

$$u_{A^c} = K_2 \tag{9}$$

In this way, both information bits and frozen bits are kept secret by using the secret key $K$. After controlling the data source $u_1^N$ by the secret key, the codeword $x_1^N$ in Eq (5) can be expressed as:

$$x_1^N = (u_A \oplus K_1) G_N(A) + K_2 G_N(A^c) \tag{10}$$

For example, we consider polar code with a code length of 8 $(N=8)$ and $P=4$. As shown in Fig 5, the secret key $K$ can be divided into $K_1$ with four white cells and $K_2$ with four gray cells. The key bits in $K_2$ are assigned in the frozen bits, while the key bits in $K_1$ are XORed with the corresponding information bits. Currently, the frozen bits in Eq (5) are the secret key bits $K_2$ rather than the whole zero sequences, and the information bits are encrypted by the secret key bit $K_1$. This completely hides the transmitted data from attackers.

As shown in Fig 4, the decoding process corresponds to the encoding process. It is assumed that the legal receiver generates the same key $K$ as the transmitter, so the same $K_1$ and $K_2$ parts are divided. $K_2$ is also stored in the frozen bits, so the SC decoder in Eq (6) can be shown as:

$$\hat{u}_i \triangleq \begin{cases} K_{2i}, & \text{if } i \in A^c \\ h_i\left(y_1^N, \hat{u}_1^{i-1}\right), & \text{if } i \in A \end{cases} \tag{11}$$

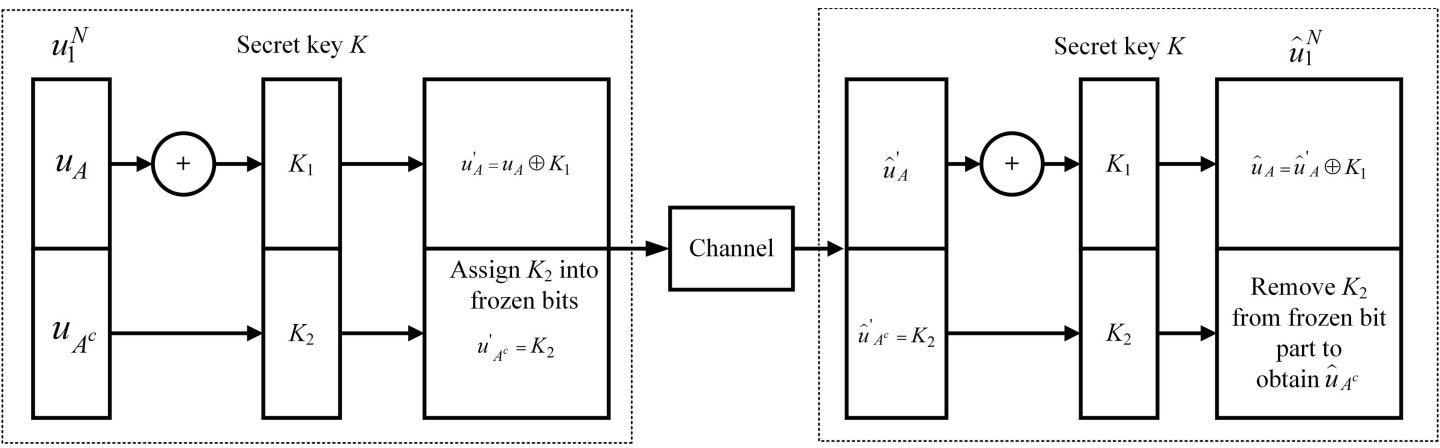

**Fig 4. Proposed crypto-coding scheme based on polar code.**

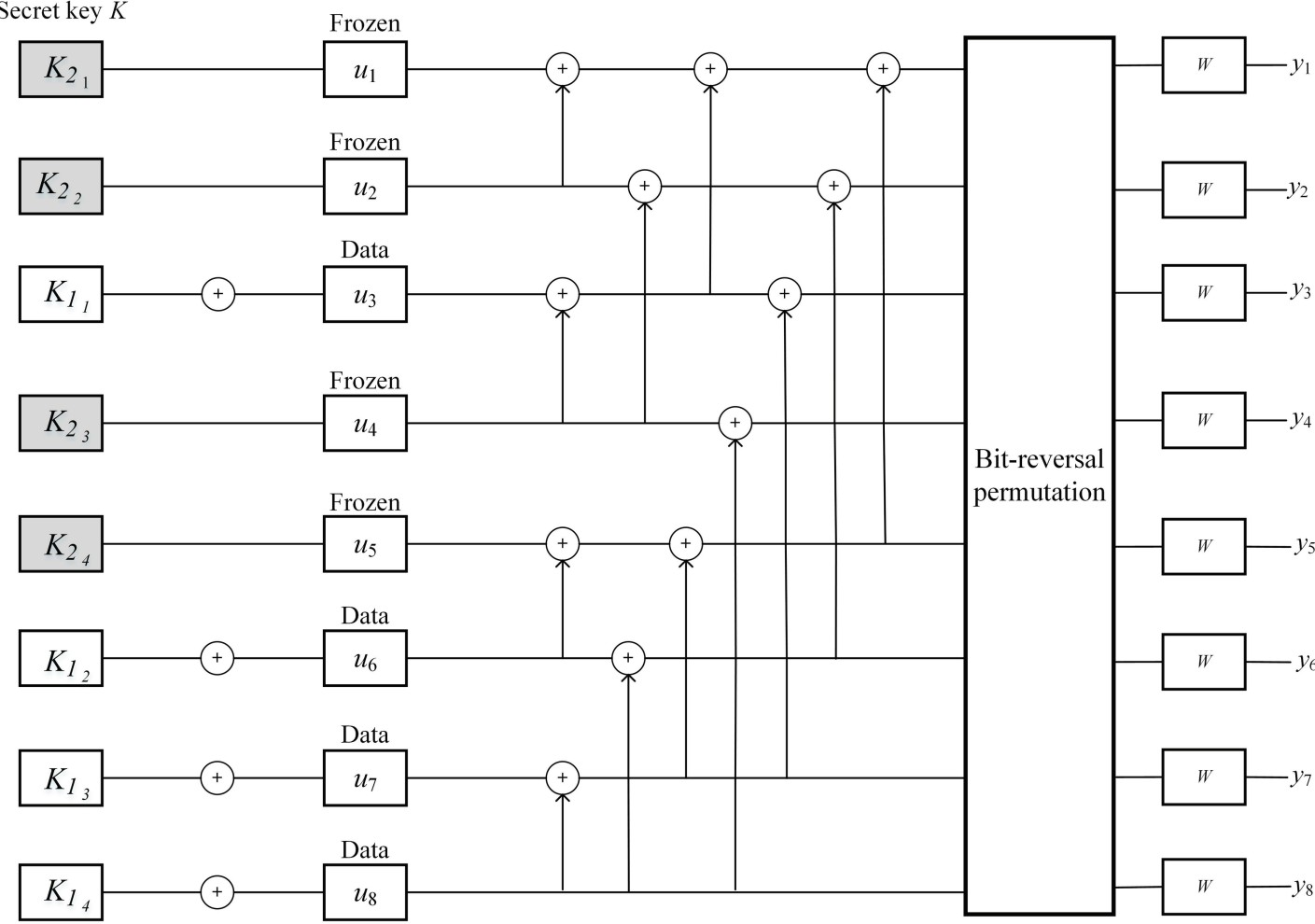

**Fig 5. An example of the proposed secure polar code.**

With $i \in A$ the received binary sequence of information part can be shown as:

$$\hat{u}'_{A_i} = h_i \left( y_1^N, \hat{u}_1^{i-1} \right) \tag{12}$$

Finally, $K_2$ will be removed from the frozen bit part and the estimated information bits $\hat{u}_A$ can be recovered by:

$$\hat{u}_A = \hat{u}'_A \oplus K_1 \tag{13}$$

The obtained data vector will be $\hat{u}_1^N = (\hat{u}_A, \hat{u}_{A^c})$.

In this proposed method, the signal is randomly processed after passing the channel coding module into the physical layer. This makes the code word after the encoding process confusing. This method not only assures error correction and encryption functions but also utilizes the physical layer's wireless channel property for secret key extraction. Consequently, this method uses a key generated from the wireless channel to control the components of the polar code, allowing it to save hardware configuration while still ensuring simultaneous implementation of error correction and encryption functions.

## III. Results and discussion

This section presents the performance of the proposed crypto-coding method in terms of error correction ability, security level, and computational complexity. To achieve this goal, we extract the secret keys from the CIR of the wireless communication systems through the AWGN channel described in Section 2 above. The randomness of these secret keys is assessed by NIST 800-22REV1A. The secret key sequence will pass the NIST test if the tested value is greater than the limit of 0.01 [36]. In our simulations, we employ the secret keys with lengths of 128 bits, 256 bits, 512 bits, 1024 bits, and 2048 bits, since these key lengths are greater than 100 bits, which can provide strong protection against brute-force attacks [37]. These keys are evaluated through 8 NIST tests [38] with some experimental parameters shown in Table 1. We need to set additional parameters of block length for NIST tests including Block frequency, Approximate entropy, and Serial. For the remaining five NIST tests, we only need to set the corresponding key length when running the results. The evaluation results are displayed in Table 2. It shows that all tested values according to NIST are greater than 0.01. Therefore, the proposed secret keys guarantee randomness.

In this research, the systems applying our proposed method are simulated by MATLAB 2020A, which is installed on a PC with Intel(R) Core(TM) i7-7500U CPU @ 2.9 GHz, 16.0 GB RAM. The simulation parameters are illustrated in Table 3.

When running the simulations, the selected key length should correspond to the code length. For example, we choose key lengths of 128, 256, 512, 1024, and 2048 bits for code

**Table 1. Parameters for some NIST tests.**

| NIST tests | Block length |
|---|---|
| Block frequency | 128 |
| Approximate entropy | 10 |
| Serial | 4 |

**Table 2. The tested values of the required NIST tests.**

| Test | 128 bits | 256 bits | 512 bits | 1024 bits | 2048 bits |
|---|---|---|---|---|---|
| Monobit | 1 | 1 | 1 | 1 | 1 |
| Block Freq. | 0.85 | 0.71 | 0.25 | 0.88 | 0.63 |
| Runs | 0.42 | 0.55 | 0.48 | 0.62 | 0.70 |
| Longest 1s | 0.49 | 0.92 | 0.63 | 0.78 | 0.38 |
| DFT | 0.76 | 0.20 | 0.36 | 0.41 | 0.28 |
| Serial 1 | 0.56 | 0.61 | 0.45 | 0.77 | 0.33 |
| Serial 2 | 0.23 | 0.78 | 0.43 | 0.52 | 0.82 |
| Approx. Entropy | 1 | 1 | 1 | 1 | 1 |
| Cum. sums | 0.98 | 0.83 | 0.72 | 0.95 | 0.85 |

**Table 3. Simulation parameters.**

| Item | Specification |
|---|---|
| Code length ($N$) | 128, 256, 512, 1024, 2048 bits |
| Key length | 128, 256, 512, 1024, 2048 bits |
| Code rate ($R$) | 1/2, 1/4, 1/8 |
| Channel | AWGN |
| Decoding algorithm | SC |
| Key length | 128, 256, 512, 1024, 2048 bits |

lengths of 128, 256, 512, 1024, and 2048 bits respectively. Additionally, for ease of representation in the figures, in different performance evaluations, we choose different key lengths/code lengths and code rates as follows:

- For evaluating the error correction ability of the proposed method:
  - In case study 1: the code lengths and key lengths of 1024 bits and 2048 bits are unchanged in each situation. Then, we modify the code rates of 1/2 and 1/4.
  - In case study 2: The code rates of 1/4 and 1/8 are unchanged in each situation. Then, we modify the code lengths and key lengths of 128 bits and 512 bits.
- For evaluating the security performance, we simulate a code length/key length of 128 bits and code rate of 1/2, and a code length/key length of 256 bits and code rate of 1/4.
- For evaluating computational complexity, we implement for all code lengths/key lengths of 128, 256, 512, 1024 and 2048 bits with a code rate of 1/2.

## Evaluation of the error correction ability of the proposed method

We simulate the systems via the AWGN channel in two case studies, including changing code rates and code lengths, to evaluate the performance of the proposed secure polar code. The BER performance system using the proposed method will be compared to that using the conventional polar code in each scenario.

*Case study 1*:

We keep code lengths and key lengths of 1024 bits and 2048 bits, then modify the code rates of 1/2, and 1/4. The BER performances of the code lengths of 1024 and 2048 are displayed in Figs 6 and 7, respectively. It can be observed that the BER of the proposed method is close to the conventional polar code. Therefore, the proposed method does not affect the BER of the system and ensures the error correction efficiency of the polar code. When the code rate $R$ increases, the BER performances of the conventional method and the proposed method decrease. This is because more channels are chosen to transmit the information bits and the bad channels will be introduced, which will raise the BER values. For example, the systems with the code length of 1024 obtain $E_b/N_0 = 2$ dB at BER = $3 \times 10^{-2}$ and $6 \times 10^{-3}$ for $R = 1/2$ and $R = 1/4$, respectively; the systems with the code length of 2048 obtain $E_b/N_0 = 2$ dB at BER around $10^{-2}$ and $10^{-3}$ for $R = 1/2$ and $R = 1/4$, respectively.

Table 4 shows the comparison of our technique with the techniques in [18], [19], and [26] for $N = 1024$ and $R = 1/2$. It can be seen that the BERs in dB of the previous studies with $E_b/N_0 = 2$ dB are around $3 \times 10^{-2}$, which are similar to those obtained by our method.

*Case study 2*:

The code rates of 1/4 and 1/8 are unchanged in each simulation and the code lengths are modified by 128 and 512 bits. The code lengths and key lengths are modified by 128 and 512 bits. The simulation results obtained in Figs 8 and 9 indicate that code length also affects the error correction ability of conventional and proposed secure polar codes. The BER performances in both cases are directly proportional to the code length. Moreover, the BER efficiency of the polar code does not change when the information bit and frozen bit are controlled by the secret key. For instance, both methods in the case of $R = 1/4$ offer BER = $10^{-3}$ with $E_b/N_0 = 3.5$ dB for $N = 256$ and 2.5 dB for $N = 512$, respectively; and both methods in the case of $R = 1/8$ obtain BER = $10^{-3}$ with $E_b/N_0 = 4$ dB for $N = 256$ and 2.75 dB for $N = 512$.

## Security performance

To assess the security level, BER curves are provided when Bob and Eve are observing Alice as shown in the system model (Fig 1). In this case, Alice sends a random message with an equal

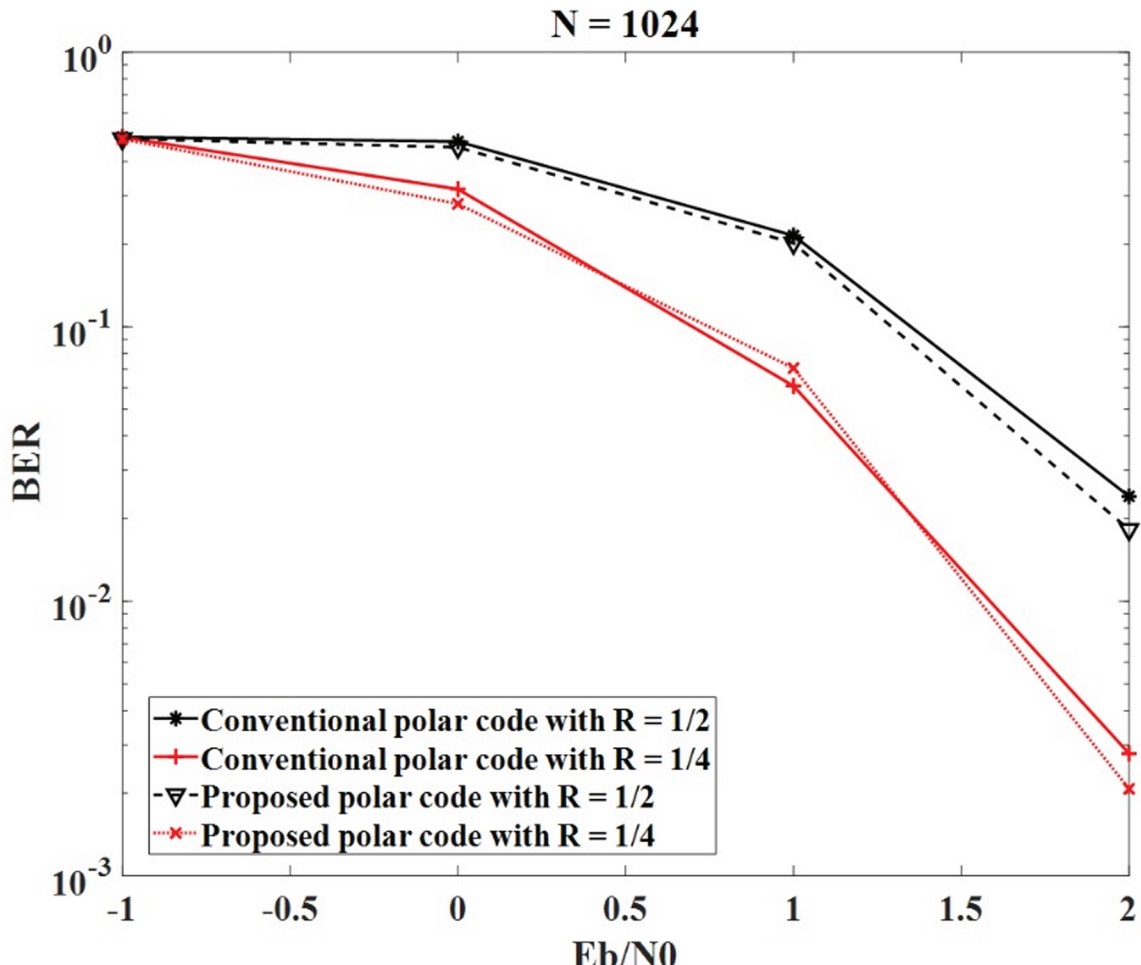

**Fig 6. The BER of the systems with conventional polar code and the proposed secure polar code in case of *N*=1024 and modified *R*.**

rate of bit 0 and bit 1, while Bob and Eve use their secret keys to decode/decrypt the received messages. Let denote the percentage of the secret key bit values unknown to the receiver. Eve obtains the secret key with $p$ values ranging from 0% to 100% ($p$ = 100%, meaning Eve's key is completely different from Alice's and Bob's keys). The $p$-value of 0% gives the BER performance of Bob in our scenario since Alice and Bob extract the same secret keys from the CIR of the systems. In the case of $p$ of 1%, only one bit or a few bits in Eve's key differs from Bob's.

In this subsection, we simulate the system applying our proposed method with different $p$ values, including 1%, 25%, 50%, and 100% for $N$ = 128 with $R$ = 1/2 and $N$ = 256 with $R$ = 1/4. The results in Figs 10 and 11 show that the BERs of various $p$ values are the lines and fluctuate from 0.3 to 1 for all ranges of $E_b/N_0$, which are completely different from the BER's Bob. This means that Eve cannot exactly decrypt the message received from Alice using his key, even if Eve's key differs by only a few bits from Bob's key. It can be concluded that our method obtains perfect secrecy.

To compare the security performance of the proposed method with the previous methods, we can employ the BER of Eve. Table 5 indicates the comparison results between our

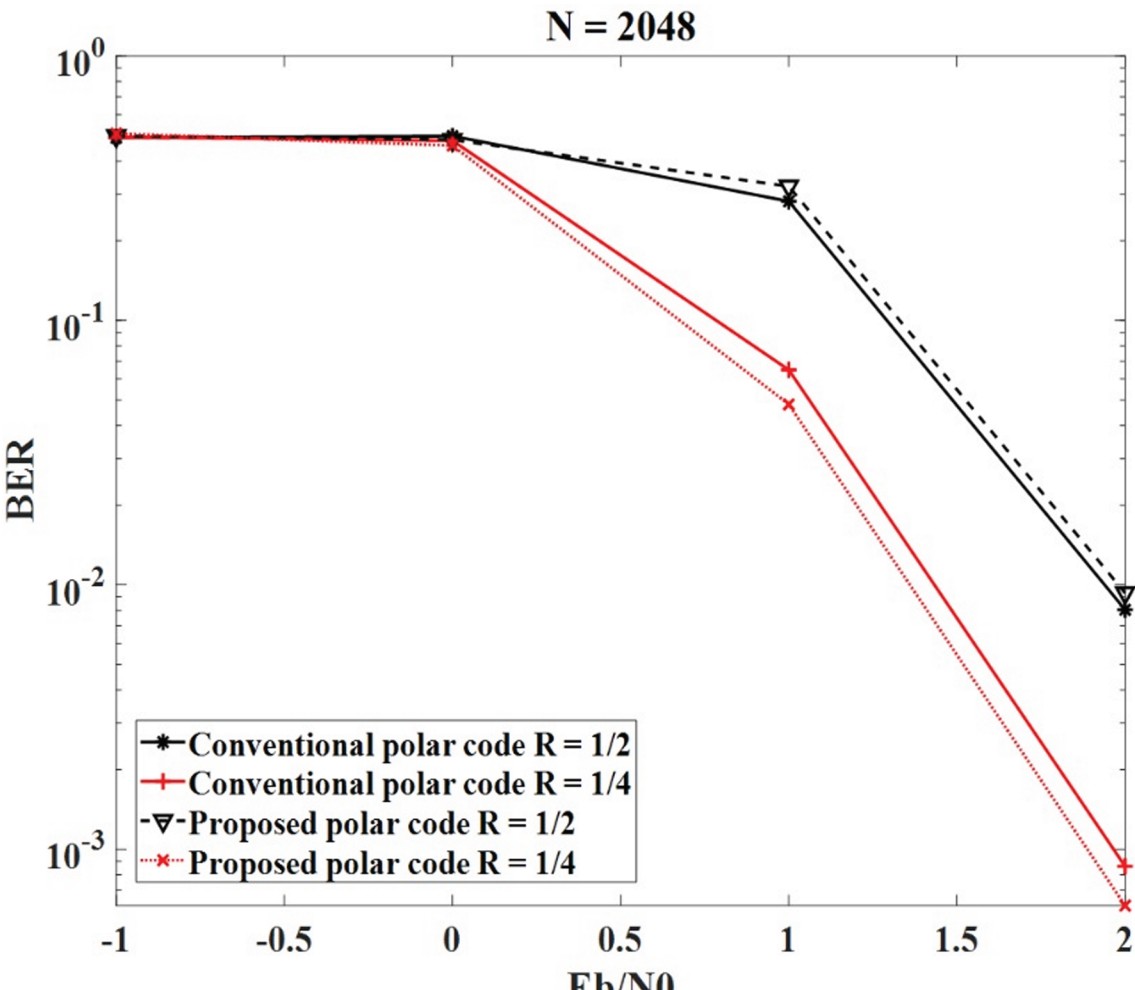

**Fig 7. The BER of the systems with conventional polar code and the proposed secure polar code in case of N=2048 and modified R.**

**Table 4. Comparing our method with the previous methods for N = 1024 and R = 1/2.**

| Techniques | BER at $E_b/N_0=2dB$ |
|---|---|
| Ours | $3 \times 10^{-2}$ |
| [18] | $\approx 3 \times 10^{-2}$ |
| [19] | $\approx 3 \times 10^{-2}$ |
| [26] | $\approx 3 \times 10^{-2}$ |

proposed method and the methods in [25] and [27]. It can be seen that our proposed method gives Eve's BER almost unchanged, and is represented on the figure as a straight line. Meanwhile, Eve's BER in studies [25] and [27] decreases with increasing $E_b/N_0$. For $E_b/N_0 = 10$ dB, BER values are 0.5 for our method, and around $3 \times 10^{-3}$ for [25] and [27], respectively. Therefore, the methods in [25] and [27] reduce Eve's decoding ability less than our method. It can be concluded that the security performance of the proposed method outperforms the existing methods in [25] and [27].

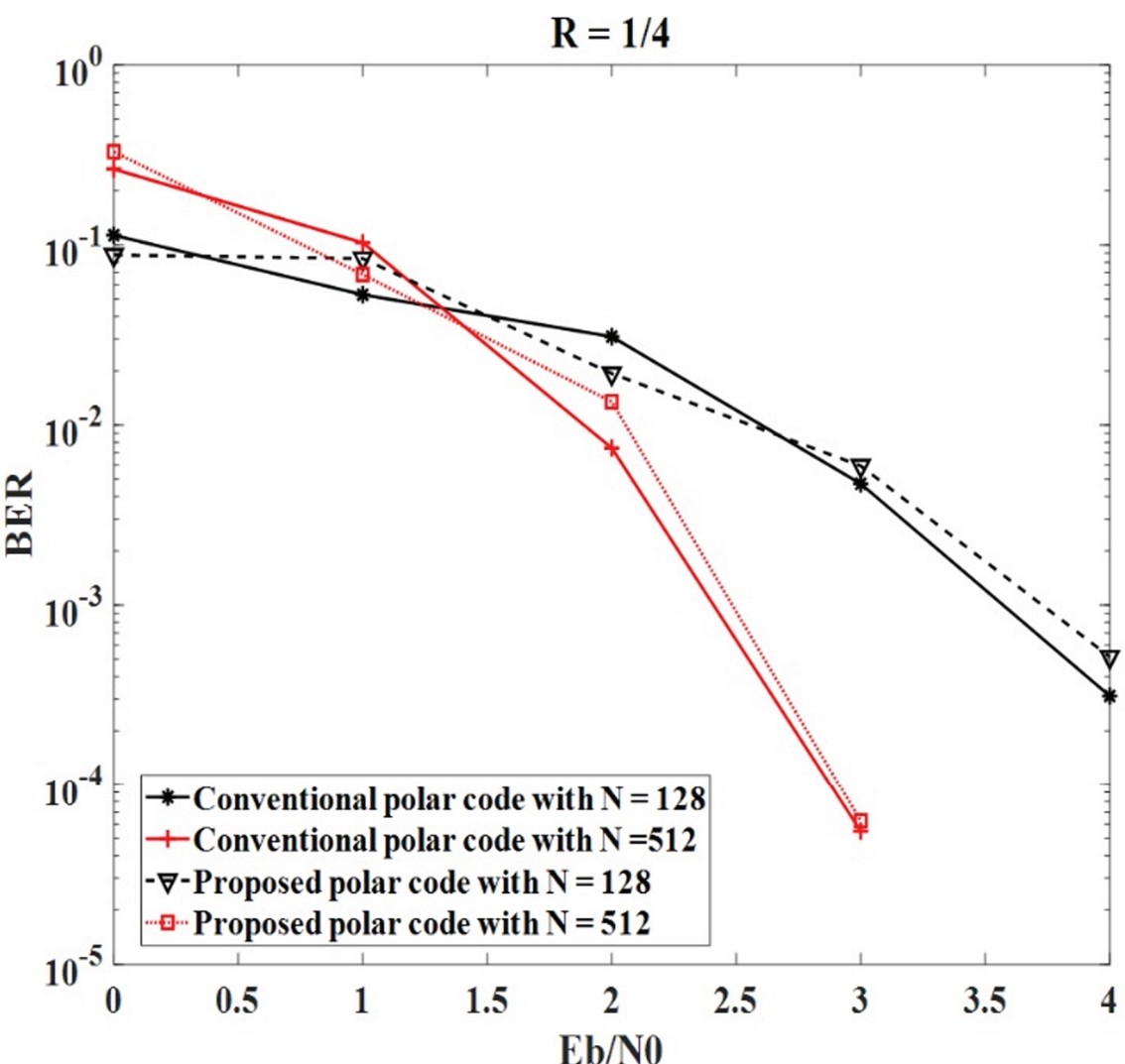

**Fig 8. The BER of the systems with conventional polar code and the proposed secure polar code in case of *R* = 1/4 and modified *N*.**

## Computational complexity

In the proposed method, there is no pre-computational stage, so the complexity of the proposed method is relevant to only the encoding and decoding procedures, which is the same as the complexity of the conventional polar code. The encoder and decoder of the proposed method can be expressed as follows [31]:

$$C_{encoding} = C_{decoding} = O(N \log N) \tag{14}$$

To prove the above statement, we calculate the time complexity of the traditional method and the proposed method using MATLAB software for *R* = 1/2. The time complexity results are shown in Table 6. It can be seen that the time complexity of our proposed method is close to that of the conventional polar code, and it increases as code length increases.

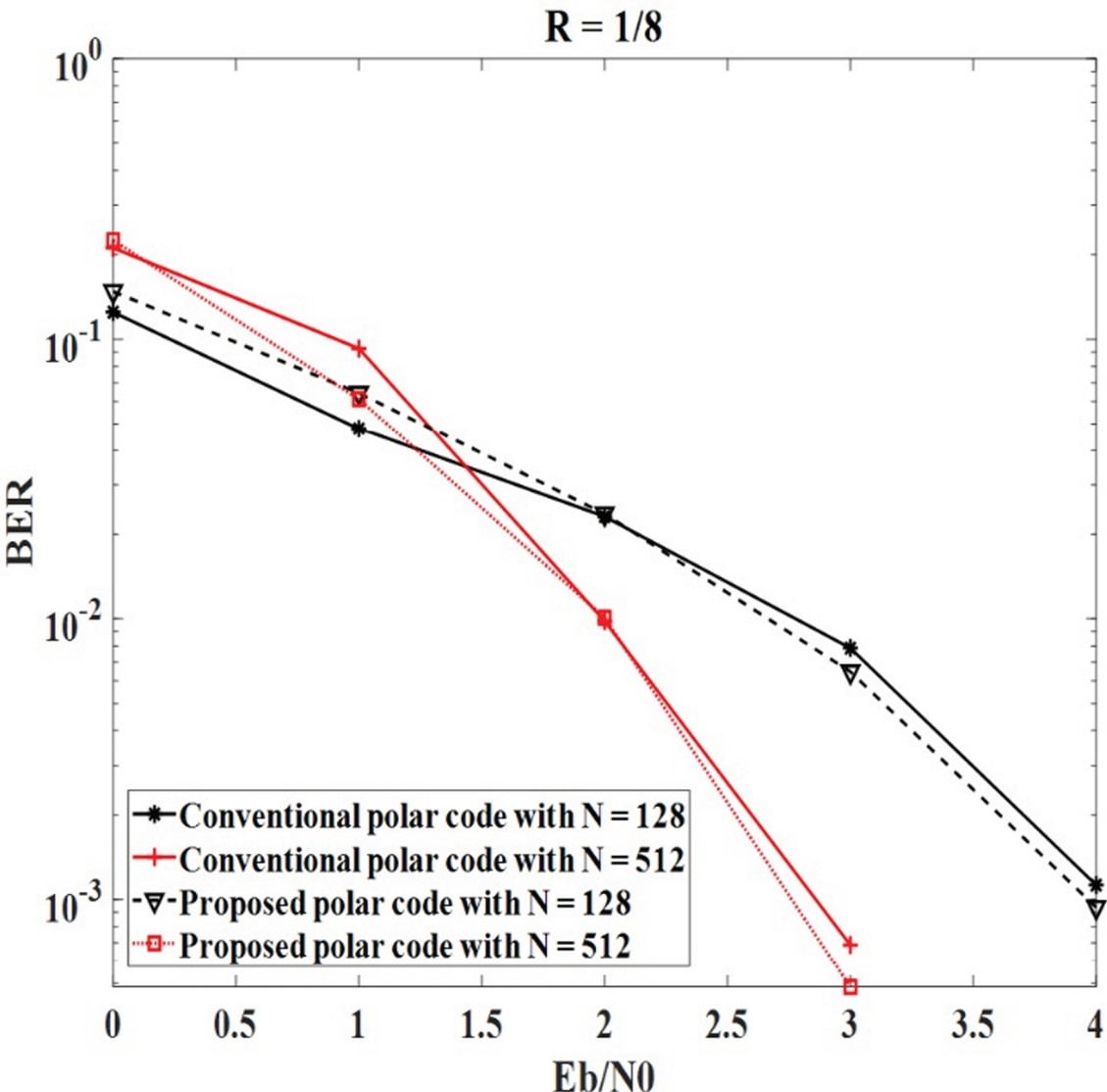

**Fig 9. The BER of the systems with the conventional polar code and the proposed secure polar code in case of *R* = 1/8 and modified *N*.**

For the computational complexity, the previous studies in [21], [39], [40] and our proposed method are similar. All methods give the complexity of $O(N \log N)$ for both encoding and decoding processes. Consequently, the computational complexity of our method is equivalent to that of the previous methods.

Based on the results, it can be concluded that our method achieves low complexity as the conventional method and the previous methods. Overall, the proposed crypto-coding does not change the error correction efficiency and computational complexity compared with the conventional polar code. In addition, it increases data confidentiality when transmitted over the wireless channel.

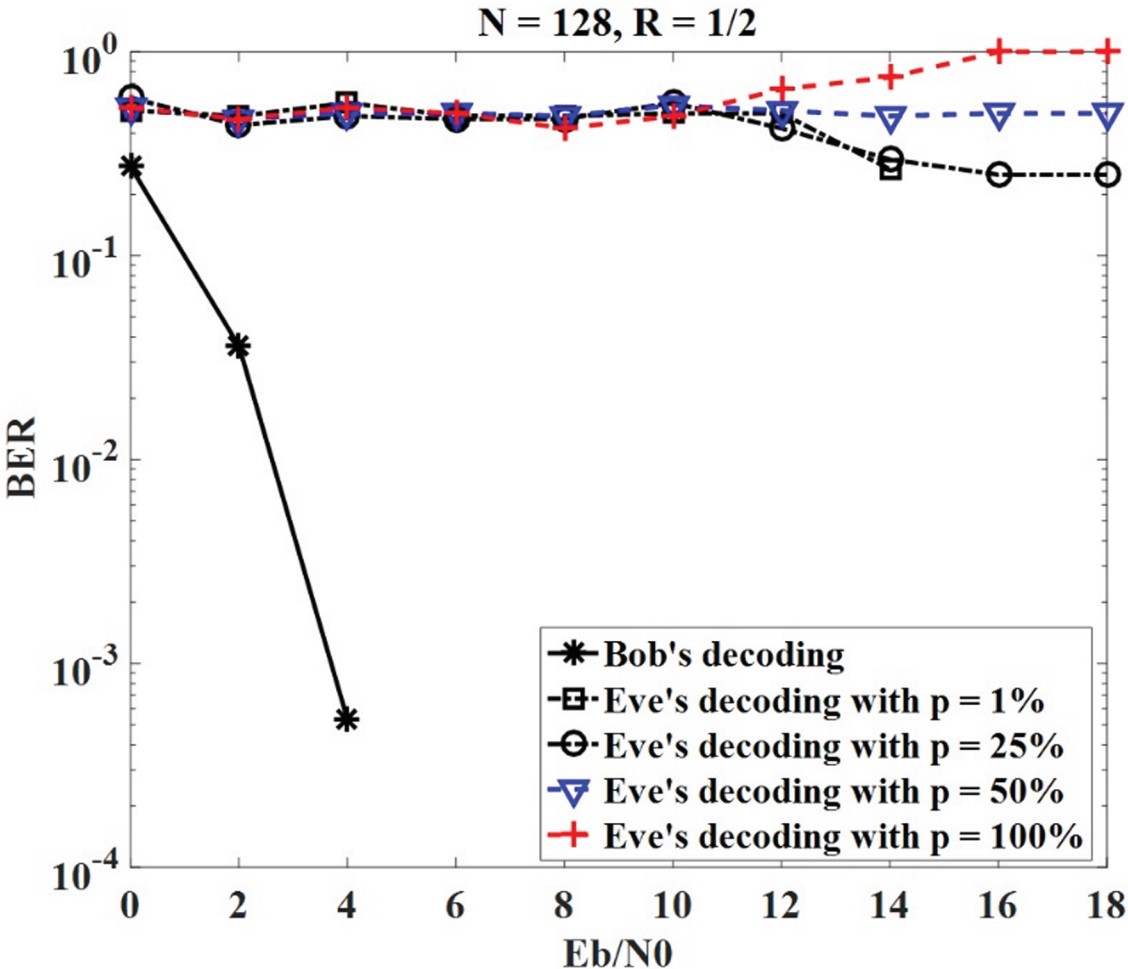

**Fig 10. Security performance for** *N* = 128, *R* = 1/2.

## IV. Conclusions

In this work, we propose a crypto-coding scheme based on the polar code and the secret key generated from the CIR for wireless communication systems. One portion of this secret key is subjected to an XOR operation with information bits, and the renaming one is assigned to frozen bits of the polar code. Our proposed secure polar code is implemented with various code lengths and code rates via the AWGN channel. The simulated results indicate that our proposed method obtains the same BER effectiveness as the traditional polar code. Our method provides the best error correction ability when using a low code rate and a great code length. The encoding and decoding processes achieve low computational complexity with $O(N \log N)$ for each. The eavesdropper cannot decrypt the cipher text at all, even if he generates a secret key that is 99% similar to the legitimate user's secret key. Moreover, this method only needs to generate a unique key from the radio channel characteristics for each communication session, so there is no requirement for a

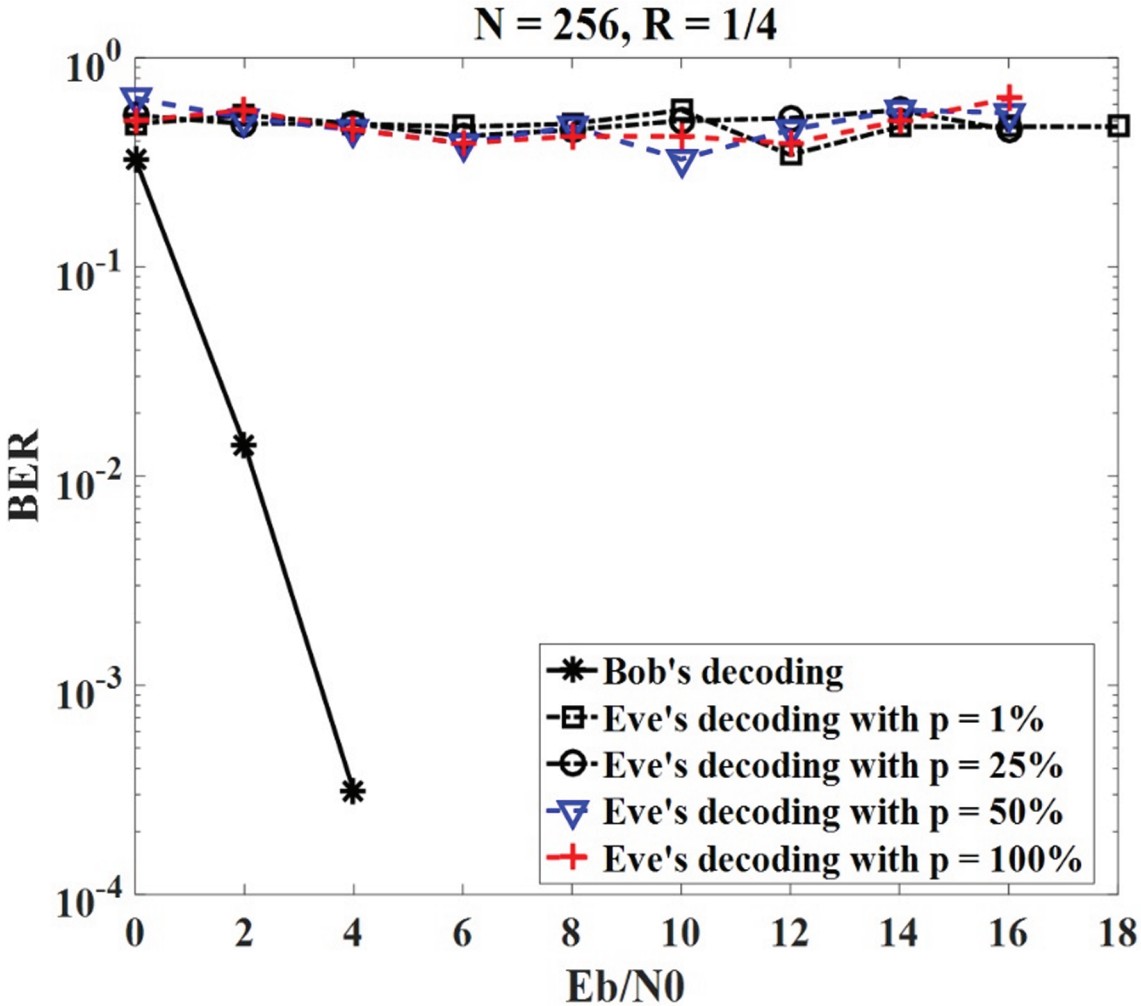

**Fig 11. Security performance for *N* = 256, *R* = 1/4.**

**Table 5. Comparing the security performance of the proposed method and the previous methods.**

| Our method | BER of Eve at $E_b/N_0$ = 10dB |
|---|---|
| Our methods | 0.5 |
| [25] | $\approx 3 \times 10^{-3}$ |
| [27] | $\approx 3 \times 10^{-3}$ |

third party to distribute and manage the key. This helps the algorithm have low complexity and does not require additional hardware structures while still ensuring error correction and security functions in the same step. Our proposed method offers a wide variety of potential applications in both military and commercial 5G communication systems. Until now, we only focused on a single-antenna channel model. In the future, we will extend this research to multiple-input multiple-output (MIMO) systems, and other advanced channel codes will be considered. Additionally, we will also implement the proposed method in the hardware structure and try more active attacks and realistic scenarios to verify its feasibility.

**Table 6. Total time complexity of the conventional polar code and the proposed secure polar code with $R = 1/2$.**

| Code length (bits) | 128 | 256 | 512 | 1024 | 2048 |
|---|---|---|---|---|---|
| Time complexity for conventional polar code | 3.43 | 9.24 | 30.12 | 116.02 | 853.54 |
| Time complexity for proposed polar code | 3.38 | 9.11 | 30.70 | 116.05 | 853.17 |

## Supporting information

**S1 Fig. The system model.**
(PDF)

**S2 Fig. Secret key generation steps for the wireless communication systems.**
(PDF)

**S3 Fig. The secure transmission model based on the polar code.**
(PDF)

**S4 Fig. Proposed crypto-coding scheme based on polar code.**
(PDF)

**S5 Fig. An example of the proposed secure polar code.**
(PDF)

**S6 Fig. The BER of the systems with conventional polar code and the proposed secure polar code in case of $N = 1024$ and modified $R$.**
(PDF)

**S7 Fig. The BER of the systems with conventional polar code and the proposed secure polar code in case of $N = 2048$ and modified $R$.**
(PDF)

**S8 Fig. The BER of the systems with conventional polar code and the proposed secure polar code in case of $R = 1/4$ and modified $N$.**
(PDF)

**S9 Fig. The BER of the systems with the conventional polar code and the proposed secure polar code in case of $R = 1/8$ and modified $N$.**
(PDF)

**S10 Fig. Security performance for $N = 128$, $R = 1/2$.**
(PDF)

**S11 Fig. Security performance for $N = 256$, $R = 1/4$.**
(PDF)

**S1 Table. Parameters for some NIST tests.**
(PDF)

**S2 Table. The tested values of the required NIST tests.**
(PDF)

**S3 Table. Simulation parameters.**
(PDF)

**S4 Table. Comparing our method with the previous methods for N = 1024 and R = 1/2.**
(PDF)

**S5 Table. CComparing the security performance of the proposed method andthe previous methods.**
(PDF)

**S6 Table. Total time complexity of the conventional polar code and the proposed secure polar code with *R* = 1/2.**
(PDF)

## Author contributions

**Conceptualization:** Dinh Van Linh.

**Data curation:** Dinh Van Linh.

**Methodology:** Dinh Van Linh, Vu Van Yem, Hoang Thi Phuong Thao.

**Software:** Dinh Van Linh.

**Supervision:** Vu Van Yem.

**Writing – original draft:** Dinh Van Linh.

**Writing – review & editing:** Hoang Thi Phuong Thao.

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
