## [Decision Letter · Decision Letter 0]

24 Jul 2024

PONE-D-24-25066Crypto-Coding Technique Based on Polar Code and Secret Key Generated from Wireless Channel Characteristics for Wireless Communication Systems.PLOS ONE

Dear Dr. Hoang Thi Phuong,

Thank you for submitting your manuscript to PLOS ONE. After careful consideration, we feel that it has merit but does not fully meet PLOS ONE’s publication criteria as it currently stands. Therefore, we invite you to submit a revised version of the manuscript that addresses the points raised during the review process.

We look forward to receiving your revised manuscript.

Kind regards,

Shahid Rahman, PhD

Academic Editor

PLOS ONE

Journal Requirements:

5. We note that your Data Availability Statement is currently as follows: All relevant data are within the manuscript and its Supporting Information files.

6. Please upload a copy of Figures 1-11, to which you refer in your text on pages 3-8. If the figure is no longer to be included as part of the submission please remove all reference to it within the text.

7. We note you have included a table to which you do not refer in the text of your manuscript. Please ensure that you refer to Table 1 and 2 in your text; if accepted, production will need this reference to link the reader to the Table.

Additional Editor Comments:

The Authors not properly follow the PLOS ONE Paper format.

The authors should rewrite the abstract in a concise form.

Literature is not presented in desirable ways.

The whole paper flow control and connectivity is ambiguous, it is difficult for readers.

Make sure the paper format in terms of Introduction, Literature, Proposed Methodology, and Results and Discussionetc.

Conduct some critical analysis to prove the strength of the proposed scheme.

Figures 4 and 5 are not well presented, blured need to explain in concise manner to show the scope of the research.

Add some comparative analysis with your method.

Clearly highlights the gaps and future work in conclusion section.

The English of the paper should be polished further.

Reviewers' comments:

Reviewer's Responses to Questions

**Comments to the Author**

1. Is the manuscript technically sound, and do the data support the conclusions?

Reviewer #1: Yes

2. Has the statistical analysis been performed appropriately and rigorously?

Reviewer #1: Yes

3. Have the authors made all data underlying the findings in their manuscript fully available?

Reviewer #1: Yes

4. Is the manuscript presented in an intelligible fashion and written in standard English?

Reviewer #1: Yes

5. Review Comments to the Author

Reviewer #1: � Please ensure that the paragraph is well-developed, coherent, concise, and can stand alone as a unit of information. It should cover all essential academic elements of the full-length paper, including the background, purpose, focus, methods, results, and conclusions.

Please review your mathematical equations once according to your paper design.

Enhance your literature by including every technique that you used in your paper.

Elaborate on the algorithm you have used. Compare your algorithm with existing ones. Compare your proposed method with existing methods.

6. PLOS authors have the option to publish the peer review history of their article (what does this mean?). If published, this will include your full peer review and any attached files.

Reviewer #1: No

---

## [Author Response · Author response to Decision Letter 1]

19 Aug 2024

Dear Dr. Shahid Rahman,

The authors would like to thank the Editor and reviewers for their helpful comments on the manuscript. Based on the reviewers' comments, the manuscript has been carefully revised from the previous submission. The detailed responses to the reviewers' comments are given in the response letter.

Please note that the page, table, and figures may have been changed in the revised manuscript.

To help with the legibility of this response letter, all the reviewers’ comments and questions are typeset in italic font and green color. Our answers are written in plain font. Rephrased sentences are typeset in blue color in this letter and are highlighted in the updated manuscript.

We are submitting two versions of the revised manuscript, a version with the

changes colored and a "clean" version.

Best regards,

Hoang Thi Phuong Thao et al.

---

## [Decision Letter · Decision Letter 1]

9 Sep 2024

PONE-D-24-25066R1Crypto-Coding Technique Based on Polar Code and Secret Key Generated from Wireless Channel Characteristics for Wireless Communication Systems.PLOS ONE

Dear Dr. Hoang Thi Phuong,

Thank you for submitting your manuscript to PLOS ONE. After careful consideration, we feel that it has merit but does not fully meet PLOS ONE’s publication criteria as it currently stands. Therefore, we invite you to submit a revised version of the manuscript that addresses the points raised during the review process.

**Please address the reviewer concerns.** Please submit your revised manuscript by Oct 24 2024 11:59PM. If you will need more time than this to complete your revisions, please reply to this message or contact the journal office at plosone@plos.org. Please include the following items when submitting your revised manuscript:

We look forward to receiving your revised manuscript.

Kind regards,

Shahid Rahman, PhD

Academic Editor

PLOS ONE

Reviewers' comments:

Reviewer's Responses to Questions

**Comments to the Author**

1. If the authors have adequately addressed your comments raised in a previous round of review and you feel that this manuscript is now acceptable for publication, you may indicate that here to bypass the “Comments to the Author” section, enter your conflict of interest statement in the “Confidential to Editor” section, and submit your "Accept" recommendation.

Reviewer #2: (No Response)

Reviewer #3: All comments have been addressed

2. Is the manuscript technically sound, and do the data support the conclusions?

Reviewer #2: Partly

Reviewer #3: Yes

3. Has the statistical analysis been performed appropriately and rigorously?

Reviewer #2: Yes

Reviewer #3: Yes

4. Have the authors made all data underlying the findings in their manuscript fully available?

Reviewer #2: Yes

Reviewer #3: Yes

5. Is the manuscript presented in an intelligible fashion and written in standard English?

Reviewer #2: Yes

Reviewer #3: Yes

6. Review Comments to the Author

**Reviewer #2: **Major corrections are needed to improve the article quality:

1. Very unsatisfied with the literature review in the Introduction section.

2. Not enough information for a reviewer to completely follow the experiments. Additional explanations to describe the experiment setup is required.

3. More technical comparisons need to be given with an existing one.

4. The article does not address other threats, such as active attacks, side-channel attacks.

5. Using wireless channel may not always be secure. What is the threat model?

6. It is not accurate to say that the method does not require additional hardware costs. Because relying on the characteristics of the wireless channel to generate and distribute keys may require additional software or firmware changes.

7. We may need to evaluate the performance of the method under more realistic scenarios to verify its practical feasibility.

8. The model has not been tested under other conditions, such as different code lengths, key lengths, or channel conditions.

**Reviewer #3: **(No Response)

7. PLOS authors have the option to publish the peer review history of their article (what does this mean?). If published, this will include your full peer review and any attached files.

Reviewer #2: No

Reviewer #3: No

---

## [Author Response · Author response to Decision Letter 2]

11 Oct 2024

To: PLOS ONE Editor

Re: Response to reviewers

Dear Editor and Reviewers,

The authors would like to thank the Editor and reviewers for their helpful comments on the manuscript. Based on the reviewers' comments, the manuscript has been carefully

revised from the previous submission. The detailed responses to the reviewers' comments

are given below.

Please note that the page, table, and figures may have been changed in the revised

manuscript.

To help with the legibility of this response letter, all the reviewers’

comments and questions are typeset in italic font and green color. Our answers are written in plain font. Rephrased sentences are typeset in blue color in this letter and are highlighted in the updated manuscript.

We are submitting two versions of the revised manuscript, a version with the

changes colored and a "clean" version.

Best regards,

Hoang Thi Phuong Thao et al.

---

## [Decision Letter · Decision Letter 2]

10 Jan 2025

Crypto-Coding Technique Based on Polar Code and Secret Key Generated from Wireless Channel Characteristics for Wireless Communication Systems.

PONE-D-24-25066R2

Dear Dr. Hoang Thi Phuong,

We’re pleased to inform you that your manuscript has been judged scientifically suitable for publication and will be formally accepted for publication once it meets all outstanding technical requirements.

Kind regards,

Shahid Rahman, PhD

Academic Editor

PLOS ONE

Additional Editor Comments (optional):

Reviewers' comments:

Reviewer's Responses to Questions

**Comments to the Author**

1. If the authors have adequately addressed your comments raised in a previous round of review and you feel that this manuscript is now acceptable for publication, you may indicate that here to bypass the “Comments to the Author” section, enter your conflict of interest statement in the “Confidential to Editor” section, and submit your "Accept" recommendation.

Reviewer #2: All comments have been addressed

2. Is the manuscript technically sound, and do the data support the conclusions?

Reviewer #2: Yes

3. Has the statistical analysis been performed appropriately and rigorously?

Reviewer #2: Yes

4. Have the authors made all data underlying the findings in their manuscript fully available?

Reviewer #2: Yes

5. Is the manuscript presented in an intelligible fashion and written in standard English?

Reviewer #2: Yes

6. Review Comments to the Author

Reviewer #2: All comments have been addressed. All comments have been addressed. All comments have been addressed.

7. PLOS authors have the option to publish the peer review history of their article (what does this mean?). If published, this will include your full peer review and any attached files.

Reviewer #2: No

---

## [Editor Report · Acceptance letter]

PONE-D-24-25066R2

PLOS ONE

Dear Dr. Thi Phuong Thao,

I'm pleased to inform you that your manuscript has been deemed suitable for publication in PLOS ONE. Congratulations! Your manuscript is now being handed over to our production team.

Kind regards,

on behalf of

Dr. Shahid Rahman

Academic Editor

PLOS ONE